# Calcium-Sensing Receptor Expression in Breast Cancer

**DOI:** 10.3390/ijms241411678

**Published:** 2023-07-20

**Authors:** Iva Busic-Pavlek, Ivo Dumic-Cule, Lucija Kovacevic, Milan Milosevic, Petra Delimar, Lea Korsa, Zlatko Marusic, Maja Prutki

**Affiliations:** 1Clinical Department of Diagnostic and Interventional Radiology, University Hospital Centre Split, Spinciceva 1, 21000 Split, Croatia; ivabusic@yahoo.com; 2Clinical Department of Diagnostic and Interventional Radiology, University Hospital Centre Zagreb, Kispaticeva 12, 10000 Zagreb, Croatia; kovacevic.lucija@gmail.com (L.K.); petrad777@gmail.com (P.D.); maja.prutki@gmail.com (M.P.); 3Department of Nursing, University North, 104 Brigade 3, 42000 Varazdin, Croatia; 4Department of Environmental Health and Occupational and Sports Medicine, Andrija Stampar School of Public Health, Rockfellerova 4, 10000 Zagreb, Croatia; milan.milosevic@snz.hr; 5School of Medicine, University of Zagreb, Salata 3, 10000 Zagreb, Croatia; 6Clinical Department of Pathology and Cytology, University Hospital Centre Zagreb, Kispaticeva 12, 10000 Zagreb, Croatia; lea.korsa@gmail.com (L.K.); marusic_zlatko@yahoo.com (Z.M.)

**Keywords:** calcium-sensing receptor, breast cancer, Ki-67 proliferation index, estrogen receptor, progesterone receptor

## Abstract

The calcium-sensing receptor (CaSR) plays a crucial role in maintaining the balance of calcium in the body. Altered signaling through the CaSR has been linked to the development of various tumors, such as colorectal and breast tumors. This retrospective study enrolled 79 patients who underwent surgical removal of invasive breast carcinoma of no special type (NST) to explore the expression of the CaSR in breast cancer. The patients were categorized based on age, tumor size, hormone receptor status, HER2 status, Ki-67 proliferation index, tumor grade, and TNM staging. Immunohistochemistry was conducted on core needle biopsy samples to assess CaSR expression. The results revealed a positive correlation between CaSR expression and tumor size, regardless of the tumor surrogate subtype (*p* = 0.001). The expression of ER exhibited a negative correlation with CaSR expression (*p* = 0.033). In contrast, a positive correlation was observed between CaSR expression and the presence of HER2 receptors (*p* = 0.002). Increased CaSR expression was significantly associated with lymph node involvement and the presence of distant metastasis (*p* = 0.001 and *p* = 0.038, respectively). CaSR values were significantly higher in the patients with increased Ki-67 (*p* = 0.042). Collectively, higher CaSR expression in breast cancer could suggest a poor prognosis and treatment outcome regardless of the breast cancer subtype.

## 1. Introduction

The calcium-sensing receptor (CaSR) is a plasma membrane receptor that is a member of the G protein-coupled receptor (GPCR) superfamily. As a GPCR, the CaSR consists of the three primary structural elements found in this family of receptors: an extracellular domain, a seven-transmembrane domain, and an intracellular tail. It was first cloned from parathyroid cells, where its expression plays a vital role in the negative feedback loop that regulates calcium homeostasis by suppressing parathyroid hormone (PTH) secretion in hypercalcemic states [1,2]. The CaSR was shown to be expressed in a plethora of diverse tissues including skeletal, renal, cardiac, hematological, ovarian, and breast tissues, where it became apparent that it is an important regulator of varied physiological processes, including the proliferation, differentiation, and apoptosis of cells.

An alteration in the signaling pathway of the CaSR has been associated with the development of a variety of tumors including colorectal and breast tumors, where the role of the CaSR has been described as that of a tumor suppressor in the former and that of an oncogene in the latter [3,4]. Normal and neoplastic breast tissues were shown to express the CaSR [5]. In the aspect of the genetic background, only few studies found a correlation between increased breast cancer risk and single-nucleotide polymorphisms (SNPs) of the CaSR gene [6]. The interaction between the CaSR and BRCA1 was analyzed, revealing that cells containing *BRCA1* mutants lacking *BRCA1* expression displayed reduced *CaSR* expression. Additionally, the findings indicated that BRCA1 utilized the CaSR to suppress the expression of survivin, a factor that promotes cell survival. Consequently, the CaSR could partially mitigate the detrimental consequences of BRCA1 loss [7].

In addition to regulating PTH, the CaSR also regulates the secretion of parathyroid hormone-related protein (PTHrP). PTHrP is actually a growth factor that utilizes the same receptors as PTH. The CaSR is expressed in normal breast epithelial cells and is activated during lactation [8]. During lactation, the CaSR enhances calcium transport into milk and participates in the regulation of systemic calcium and bone metabolism. In breast tissue cells during lactation, the CaSR suppresses the production of PTHrP. As mentioned earlier, PTHrP is a growth factor that affects calcium homeostasis in the body. In the mother’s systemic circulation, PTHrP activates a mechanism of bone resorption to increase the availability of calcium for milk production [9]. In the child’s circulation, PTHrP, through mechanisms that are not yet fully understood, influences calcium accumulation in the bones [10].

Further research has also indicated the involvement of the CaSR in a wide range of processes such as cell proliferation, cell differentiation, apoptosis, hormone secretion, and gene expression. The CaSR has been found in breast cancer cell cultures, and it has been shown that the expression of this receptor is directly associated with the occurrence of bone metastases. Unlike the physiological effect of the CaSR in suppressing the secretion of PTHrP, in breast cancer cells, the CaSR acts to stimulate the production of PTHrP. The secretion of PTHrP leads to bone resorption and an increase in the systemic concentration of Ca^2+^ ions. Elevated levels of calcium ions in breast cancer cells then promote the production of PTHrP, likely through a mechanism mediated by the CaSR. Increased levels of PTHrP, in turn, have osteolytic effects, releasing a new amount of calcium ions and establishing a positive feedback mechanism that further promotes massive osteolysis.

The CaSR has more recently been studied as a hypothetical predictive marker for skeletal metastases in breast carcinoma, and it was shown that in patients with advanced, metastatic breast cancer, CaSR expression was higher in those with skeletal metastases [11]. Based on recent findings, we aimed to explore the correlation between CaSR expression and different pathohistological prognostic factors of breast cancer. Furthermore, we compared CaSR expression with the value of the Ki-67 proliferation index, which serves as a marker of active cell proliferation and clearly indicates the biological aggressiveness of cancer.

## 2. Results

A total of 79 female patients with breast cancer of NST were included in this retrospective study. The mean age of the patients was 56.8 years (range: 28–79 years). Other clinical parameters and pathological findings of the enrolled patients are presented in Table 1.

A positive correlation was found between the expression of the CaSR and tumor size, regardless of the tumor type (*p* = 0.001) (Table 2). The expression of ER, as a hormone-dependent receptor, exhibited a negative correlation with CaSR expression (*p* = 0.033) (Table 2). In contrast to ER, a positive correlation was observed in relation to the HER2 receptor (*p* = 0.002). Increased expression of the CaSR was significantly associated with lymph node involvement and the presence of distant metastasis (*p* = 0.001 and *p* = 0.038) (Table 2).

Differences in CaSR values in breast cancer regarding the assessment of breast cancer biological aggressiveness based on the level of the Ki-67 proliferation index were observed. CaSR values were significantly higher in the Ki-67 group with values > 20: 3.5 (2.0–4.0) compared to 1.0 (1.0–5.0); *p* = 0.042 (Table 3 and Figure 1). Through ROC analysis of CaSR values in breast cancer for evaluating the biological aggressiveness of breast cancer based on the level of the Ki-67 proliferation index > 20, an optimal cutoff value of CaSR > 1 was determined with the best combination of sensitivity (83.3%) and specificity (57.89%) (Figure 2).

## 3. Discussion

In our study, which was a retrospective analysis of 79 patients with NST invasive breast cancer, the CaSR was found to be a relevant marker of tumor size and aggressiveness, irrespective of the tumor surrogate subtype. Moreover, elevated CaSR expression was significantly linked to lymph node involvement and the presence of distant metastasis. A positive correlation was noticed between CaSR expression and the presence of HER2 receptors, while the patients with elevated Ki-67 exhibited significantly higher CaSR values.

In breast cancer cells, the CaSR acts as an oncogene and promotes tumor growth through mechanisms that are not yet fully understood. Studies on mice and breast cancer cell cultures have shown that inhibition of the CaSR reduced the proliferation of breast cancer cells, and in mice with CaSR inhibition, there was slower tumor growth and longer survival compared to the control group [12,13]. Although not all mechanisms by which the CaSR affects tumor growth have been clarified, our study showed a significant positive correlation between tumor size and the expression of the CaSR. Mice with inhibited CaSR in breast cancer cells exhibited slower tumor growth and longer survival compared to the control group. VanHouten’s research on the expression level of the CaSR in metastatic breast cancer indicated a positive association between CaSR expression and lymph node involvement, as well as a negative association with progesterone receptor expression. However, we did not find any correlation between CaSR and progesterone receptor expression. All of the mentioned studies highlight the important role of the CaSR in the development and progression of breast cancer [14,15]. The activation of an expressed CaSR on two human breast cancer lines, MDA-MB-231 and MCF-7, led to increased production of parathyroid hormone-related protein (PTHrP). The secretion of PTHrP by neoplastic cells can activate PTH receptors in osteoblasts, thus activating a cascade of events that results in osteoclast-led osteolysis and further proliferation of cancerous cells [14,15]. PTHrP exerts its action through osteoblasts, by activating the RANK–RANKL–OPG system. In this activation loop, RANKL binds to the RANK receptor on osteoclasts and stimulates osteoclastogenesis [16,17]. PTHrP expression has been implicated as a risk factor for the development of skeletal metastases, in which it is more commonly expressed when compared to primary breast carcinomas [18,19]. The role of the CaSR in the development of bone metastases has already been described in breast cancer cells [14,20]. Unlike in physiological conditions where the CaSR acts to reduce bone degradation in situations of increased Ca^2+^ levels, in breast cancer cells, the CaSR acts to promote further bone resorption and an elevated systemic concentration of Ca^2+^ ions in response to an increase in the Ca^2+^ ion concentration [21]. This establishes a mechanism of positive feedback that promotes further massive osteolysis [22]. The CaSR has been linked to the development of bone grafts in research and positively correlates with their size and occurrence. In vivo studies have shown that overexpression of the CaSR in the MDA-MB-231 breast cancer cell line increases osteolytic potential by increasing osteoclastogenesis [23]. An increase in the number of osteoclasts results in increased bone resorption, which subsequently enables faster growth of tumor grafts [24]. Activation of the calcium receptor stimulates the proliferation of osteoclasts by stimulating PTHrP, which acts as a growth factor, as previously described [25,26]. Our study observed a significant, positive correlation between the expression of the CaSR and the development of distant metastases, correlating with the results of previous studies.

Considering the described characteristics of breast cancer with high CaSR expression, it is not surprising that statistical analysis confirmed a significant positive correlation between CaSR values in breast cancer and the Ki-67 proliferation index, which serves as a marker of the biological aggressiveness of breast cancer. Our study found that larger tumors with positive lymph nodes and distant metastases at the initial presentation had higher levels of the CaSR. The described characteristics, as well as Ki-67 values, indicate tumor aggressiveness. CaSR values were significantly higher in the Ki-67 group with values greater than 20 compared to patients with Ki-67 values less than 20. Within healthy breast tissue, Ki-67 can be detected in cells that do not express ER, while cells with estrogen receptors do not exhibit Ki-67 [27]. Since the expression of the CaSR significantly negatively correlates with ER expression, it is possible that ER also plays a role in the relationship with Ki-67. The results of this study correlate with findings in the literature showing that tumors with a higher malignant potential exhibit higher Ki-67 values, higher CaSR levels, and morphological characteristics associated with more malignant lesions [28,29]. Further research on larger tumors could confirm the existence of this correlation.

Approximately 70% of breast cancers are ER-positive and belong to the group of hormone-dependent tumors. The impact of estrogen in breast cancer development has already been established, with findings indicating that patients with high expression of estrogen receptor (ER) have a more favorable prognosis compared to those with ER-negative tumors, which tend to be more aggressive and prone to metastasis [30]. On the other hand, in physiological conditions, the presence of estrogen receptors is important for maintaining bone mass. In postmenopausal women and women undergoing tamoxifen therapy, which selectively acts on estrogen receptors and reduces estrogen binding to the receptor, a significant decrease in total bone mass has been observed. This reduction in bone mass is partly explained by increased osteoclast activity in the absence of estrogen. In the case of breast cancer, when the presence of the CaSR through positive feedback mechanisms leads to increased bone resorption, it has been shown that there is downregulation and decreased expression of ER receptors [31]. The mechanism by which the CaSR downregulates ER is not fully understood, but the literature indicates that high extracellular Ca^2+^ levels affect ER transcriptional activity in MCF-7 breast cancer cell lines [32]. Nevertheless, it is possible that the release of Ca^2+^ mediated by PTHrP through the CaSR influences ER regulation. This study supports the significant negative correlation between the presence of the CaSR and ER expression.

Prior to the invasion of cancer cells into the circulation, they need to undergo a process of epithelial–mesenchymal transition (EMT). In this transformation, in situ microcalcifications composed of calcium oxalate and hydroxyapatite play an important role [33,34]. The occurrence of hydroxyapatite, a calcium mineral, is associated with malignant lesions [35]. These studies indicate a significant role of calcium signaling in the spread of breast cancer. Through the previously described positive feedback loop with Ca^2+^, as well as the described influence of the CaSR on epithelial–mesenchymal transition, tumor spread is facilitated. These mechanisms can explain the positive correlation between the CaSR and the spread of breast cancer to lymph nodes and distant sites.

## 4. Materials and Methods

### 4.1. Patients

This single-center retrospective study was approved by the institutional review board, and the need for informed consent was waived. Patients who underwent surgical resection of breast cancer were enrolled in this study. Demographic, clinical, and pathological data were collected from the institutional database. Histological tumor types were classified according to the World Health Organization Histological Classification of Breast Tumors. Tumor grading was assessed according to the Elston and Ellis criteria. Only patients with invasive breast cancer of no special type (NST) were included in this study. A total of 79 patients with breast cancer of NST were selected and categorized, according to the age of the patients, size of the tumor, ER, PR, and HER2 status, Ki-67 proliferation index, histological grade of the tumor, lymphovascular invasion, lymph node status, and TNM staging of the breast cancer according to the American Joint Committee on Cancer (AJCC) 8th edition TNM system [36].

### 4.2. Immunohistochemistry

ER, PR, HER2, and Ki-67 statuses were determined through immunohistochemistry (IHC) analyses with streptavidin-peroxidase detection by staining formalin-fixed, paraffin-embedded, 3 μm thick tissue sections representative of the tumor. The ER or PR status was positive when at least 1% of the tumor cell nuclei showed staining for ER or PR, according to the Breast Biomarker Reporting guidelines of the College of American Pathologists (CAP). The HER2 status was determined positive when the IHC staining intensity score was greater than or equal to three (circumferential membrane staining that is complete, intense, and within >10% of tumor cells) according to the CAP guideline recommendations for HER2 testing in breast cancer. The determination of a HER2/CEP17 ratio ≥ 2.0 and an average HER2 copy number ≥ 4.0 via silver in situ hybridization (SISH) is considered to indicate a positive HER2 status. Surrogate definitions based on immunohistochemical analysis of breast cancer tissue were used, and subtypes were determined based on the receptor status as luminal A like (ER+ and PR+, HER2−, Ki-67 < 20%), luminal B HER2+ like (ER+ and/or PR+, HER2+, Ki-67 > 20%), luminal B HER2 negative like (ER+ and/or PR+, HER2−, Ki-67 > 20%), HER2 positive (ER−, PR−, HER2+), and triple negative or basal like (ER−, PR−, HER2−).

CaSR IHC was performed on core needle biopsy samples by staining 3 μm thick, formalin-fixed, paraffin-embedded tissue sections representative of the tumor using an automatic immunostainer, Ventana BenchMark ULTRA, Roche Diagnostics. An anti-CaSR polyclonal antibody, PA1-934A (AffinityBioReagents, Inc., Golden, CO, USA, Thermo Scientific Inc., Rockford, IL, USA; dilution 1:200), was used as a primary antibody. Evaluation of the immunohistochemical analysis of CaSR reactivity was performed in consensus by two pathologists who were blinded to other information. Expression of the CaSR was quantified according to a 6-point scale, ranging from score 0 (negative) to score 5 (strong, uniform expression), as described in the literature [11]. The expression of the CaSR was quantified as absent expression (0), rare positive cells (1), non-uniform weak expression (2), non-uniform weak/intense expression (3), intense non-uniform expression (4), or strong uniform expression (5) (Figure 2).

### 4.3. Statistical Analysis

For CaSR expression, we used two main categories: CaSR positive if the score was 3–5, and CaSR negative if the score was 0, 1, or 2. Differences in continuous data between CaSR groups were compared with the Mann–Whitney U test. Spearman rho correlation coefficients were used to assess correlations between CaSR expression and other clinical variables. ROC analysis of CaSR values in breast cancer for assessing the biological aggressiveness of breast cancer based on the level of the Ki-67 proliferation index > 20 was carried out. All *p*-values below 0.05 were considered significant. IBM SPSS statistical package for Windows, version 29.0 was used in all statistical procedures.

CaSR expression (positive versus negative) among groups was evaluated using Fisher’s exact test (two proportions) or a chi-square test (more than two proportions). The Kruskal–Wallis test was used for the comparison of expression scores among different groups. Values of *p* less than 0.05 were considered statistically significant.

## 5. Conclusions

Collectively, the results of this study indicate that there is a positive relationship between CaSR expression and tumor size, irrespective of the tumor surrogate subtype. Moreover, the expression of ER demonstrates a negative correlation with CaSR expression. Conversely, a positive correlation is observed between CaSR expression and the presence of HER2 receptors. Additionally, elevated CaSR expression is significantly associated with lymph node involvement and the presence of distant metastasis. Furthermore, patients with increased Ki-67 exhibit significantly higher CaSR values. Overall, these results suggest that higher CaSR expression in breast cancer could indicate a poor prognosis and treatment outcome, regardless of the subtype of breast cancer.

## Figures and Tables

**Figure 1 ijms-24-11678-f001:**
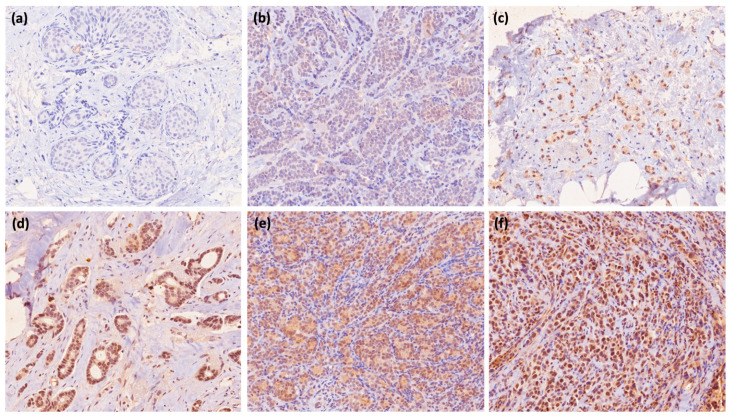
Immunohistochemical analysis of calcium-sensing receptor staining (CaSR) in breast cancer: (**a**) no expression (0), 20×; (**b**) rare positive cells (1), 20×; (**c**) non-uniform weak expression (2), 20×; (**d**) non-uniform weak/intense expression (3), 20×; (**e**) non-uniform intense expression (4), 20×; (**f**) strong uniform expression (5), 20×.

**Figure 2 ijms-24-11678-f002:**
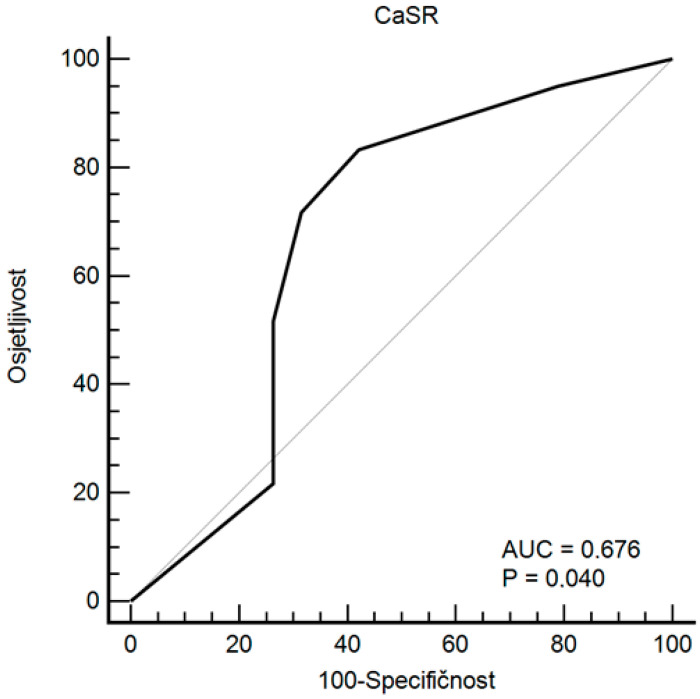
ROC analysis of CaSR values in breast cancer for assessing the biological aggressiveness of breast cancer based on the level of the Ki-67 proliferation index > 20.

**Table 1 ijms-24-11678-t001:** Descriptive statistics of the clinical categorical variables among the enrolled patients. ER—estrogen receptor; PR—progesterone receptor.

		N	%
Affected side	Left	37	46.8%
	Right	42	53.2%
Tumor type	Unifocal	57	72.2%
	Multicentric	16	20.3%
	Multifocal	6	7.6%
Local invasion	No invasion	59	74.7%
	Skin infiltration	7	8.9%
	Infiltration of mamilla	6	7.6%
	Pectoral muscle infiltration	7	8.9%
	Combination	0	0.0%
ER	Negative	10	12.7%
	Positive	69	87.3%
PR	Negative	18	22.8%
	Positive	61	77.2%
HER	Negative	64	81.0%
	Positive	15	19.0%
Ki-67	≤20	19	24.1%
	>20	60	75.9%

**Table 2 ijms-24-11678-t002:** Correlation between CaSR expression and prognostic factors in breast cancer. ER—estrogen receptor; PR—progesterone receptor; LVI—lymphovascular invasion; LNI—lymph node involvement; T—tumor; N—lymph node; M—metastasis.

		CaSR
Age (years)	Correlation coefficient	−0.076
	*p*	0.503
	N	79
Tumor diameter (mm)	Correlation coefficient	0.379
	*p*	0.001
	N	79
ER (%)	Correlation coefficient	−0.240
	*p*	0.033
	N	79
PR (%)	Correlation coefficient	−0.114
	*p*	0.316
	N	79
HER	Correlation coefficient	0.340
	*p*	0.002
	N	79
Ki-67 (value)	Correlation coefficient	0.204
	*p*	0.072
	N	79
Grade	Correlation coefficient	0.105
	*p*	0.356
	N	79
LVI	Correlation coefficient	0.217
	*p*	0.055
	N	79
LNI	Correlation coefficient	0.383
	*p*	<0.001
	N	79
T	Correlation coefficient	0.214
	*p*	0.059
	N	79
N	Correlation coefficient	0.378
	*p*	0.001
	N	79
M	Correlation coefficient	0.234
	*p*	0.038
	N	79

**Table 3 ijms-24-11678-t003:** Differences in CaSR values in the aspect of breast cancer biological aggressiveness based on the level of the Ki-67 proliferation index.

Ki-67		N	Mean	SD	Min.	Max.	Centile			*p*
							25.	Median	75.	
CaSR	≤20	19	2.16	2.01	0.00	5.00	1.00	1.00	5.00	0.042
	>20	60	3.20	1.48	0.00	5.00	2.00	3.50	4.00	

## Data Availability

Data used for analysis are contained within the article.

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
