# Peer review of "Calcium-Sensing Receptor Expression in Breast Cancer"

_ijms, 2023, doi:10.3390/ijms241411678_

Round 1

Reviewer 1 Report

This article is covering calcium-sensing receptor (CaSR) role in maintaining the balances of calcium in the body and its link to the development of tumors including breast cancer. The authors have studied the CaSR expression and its correlation with tumor size. They concluded that the expression of the estrogen receptor (ER) exhibited a negative correlation with CaSR expression. Higher CaSR expression in breast cancer was linked to and suggests its poor prognosis in available treatment therapy.  

Additionally, article is compiling important data on correlation between CaSR expression and prognostic factors in breast cancer including estrogen receptor (ER), progesterone receptor (PR), lymphovascular invasion (LVI), lymph node involvement (LNI), herceptin receptors (HER2), proliferation index value (Ki-67), and age of the patient and tumor size.

Particularly informative is Table 2, clearly characterizing important correlation (positive and negative) related to CaSR values. The authors should be complimented for such an excellent summary of correlative analysis of prognostic factors in breast cancer. This will constitute crucially important goals and novelty of this very important paper.

The following suggested small changes and recommendations should be introduced before the publication of the manuscript.

1.     Page 2, Line 83.Insert “clearly” before “indicates”

2.     Page 3. Table 2. Third column. CASR should be corrected to CaSR. 

3.     Page 4  Line 101,105 and 108. Insert hyphen in Ki67 and correct to Ki-67.

4.     Page 6  Line 145 and all text of the manuscript.  Ca2+, should be in superscript  “Ca2+ “.

The manuscript is of very good quality and urgent importance and is well written and edited in order to meet the standard for the articles published in International Journal of Molecular Sciences. Thus, I certainly recommend it for publication after the correction of these suggested minorchanges. 

Author Response

We thank this reviewer for the constructive criticism which has improved the quality of our revised manuscript.

Following suggestions are corrected, corrections are marked with red color:

1)   Page 2, Line 83. Insert “clearly” before “indicates”

2)   Page 3. Table 2. Third column. CASR should be corrected to CaSR.

3)   Page 4  Line 101,105 and 108. Insert hyphen in Ki67 and correct to Ki-67.

4)   Page 6  Line 145 and all text of the manuscript.  Ca2+, should be in superscript  “Ca2+ “.

Reviewer 2 Report

The submitted manuscript describes the results of the retrospective study on patients who underwent surgical removal of invasive breast carcinoma to explore the expression of CaSR in breast cancer. The study is quite simple and short, I wouldn’t call it a “Full article” but rather a Communication. Nevertheless, despite the limited amount of methods and results, the study is quite interesting. Therefor I recommend the revision, including the points listed below.

Line 37, CaSR acronym must be defined here, even if it was defined in the abstract

Lines 37-38, some more information on the structure of CaSR should be presented here

Line 72 and further, in the discussion; it should be Ca2+

Table 1 caption, acronyms used in the table must be defined in its caption

Table 2, again, some symbols (T, N, M) were not defined. Of course, I guess what they mean but it should be stated clearly.

Lines 128-131, a reference to this study is needed here

I really miss some graphs, showing i.e. the dependance on Ki from CaSR expression

Lines 177-179, actually, the role of estrogen receptor is more significant and includes many other aspects

Line 196, “previously” appears twice in the same line

The authors have measured quite a few parameters but the applied statistics is rather simple. I recommend improving the statistical analysis by using the chemometric methods. This can lead to more significant conclusions.

Conclusions section is missing. It does not need to be long, but it should be included.

Author Response

We thank this reviewer for the constructive criticism which has improved the quality of our revised manuscript.

Following suggestions are corrected, corrections are marked with red color:

Line 37, CaSR acronym must be defined here, even if it was defined in the abstract

Lines 37-38, some more information on the structure of CaSR should be presented here

Line 72 and further, in the discussion; it should be Ca2+

Table 1 caption, acronyms used in the table must be defined in its caption

Table 2, again, some symbols (T, N, M) were not defined. Of course, I guess what they mean but it should be stated clearly.

Lines 128-131, a reference to this study is needed here

Line 196, “previously” appears twice in the same line

According to reviewer suggestion we included conclusion section.

Reviewer 3 Report

A study "The Calcium Sensing Receptor Expression in Breast Cancer" by Iva Busic-Pavlek et al retrospectively investigates potential clinical significance of CaSR in breast cancer. The study is properly designed and the manuscript is concisely but clearly written.

Specific comments:

1. Gene names should be written in italics.

2. Please provide consistently written abbreviation for CaSR throughout the manuscript.

3. The conclusions should be added.

4. Figure 2 should be shown in the result section of the manuscript.

Author Response

We thank this reviewer for the constructive criticism which has raised important questions and significantly improved the quality of our revised manuscript. Reviewers comments are marked in text with red color.

1) Gene names should be written in italics.

2) Please provide consistently written abbreviation for CaSR throughout the manuscript.

3) The conclusions should be added.

4) Figure 2 should be shown in the result section of the manuscript.

Round 2

Reviewer 2 Report

The Authors have revised their manuscript, following my suggestions. This version can be accepted for publication.